# Corticospinal Excitability to the Biceps Brachii is Not Different When Arm Cycling at a Self-Selected or Fixed Cadence

**DOI:** 10.3390/brainsci9020041

**Published:** 2019-02-14

**Authors:** Evan J. Lockyer, Anna P. Nippard, Kaitlyn Kean, Nicole Hollohan, Duane C. Button, Kevin E. Power

**Affiliations:** 1Faculty of Medicine, Memorial University of Newfoundland, St. John’s, NL A1C5S7, Canada; ejl006@mun.ca (E.J.L.); dbutton@mun.ca (D.C.B.); 2School of Human Kinetics and Recreation, Memorial University of Newfoundland, St. John’s, NL A1C5S7, Canada; apn688@mun.ca (A.P.N.); kmk020@mun.ca (K.K.); neh420@mun.ca (N.H.)

**Keywords:** motor evoked potential, MEP, arm cranking, pedalling, exercise

## Abstract

Background: The present study compared corticospinal excitability to the biceps brachii muscle during arm cycling at a self-selected and a fixed cadence (SSC and FC, respectively). We hypothesized that corticospinal excitability would not be different between the two conditions. Methods: The SSC was initially performed and the cycling cadence was recorded every 5 s for one minute. The average cadence of the SSC cycling trial was then used as a target for the FC of cycling that the participants were instructed to maintain. The motor evoked potentials (MEPs) elicited via transcranial magnetic stimulation (TMS) of the motor cortex were recorded from the biceps brachii during each trial of SSC and FC arm cycling. Results: Corticospinal excitability, as assessed via normalized MEP amplitudes (MEPs were made relative to a maximal compound muscle action potential), was not different between groups. Conclusions: Focusing on maintaining a fixed cadence during arm cycling does not influence corticospinal excitability, as assessed via TMS-evoked MEPs.

## 1. Introduction

It is well established that rhythmic locomotor outputs in non-human animals (e.g., cat, rat, and dog) are partially controlled by neural circuits located in the spinal cord, referred to as central pattern generators (CPGs) [1,2]. Evidence, albeit indirect, has shown that the CPGs also contribute to the production of rhythmic motor outputs in humans by integrating descending and afferent inputs [3,4]; though it is believed that the descending input is of greater importance in the control of human locomotor outputs [5].

Arm cycling has been introduced as a model of locomotor output for examining changes in neural excitability during rhythmic movement, with the vast majority of these studies using a set cadence and power output for each participant [4,5]. While this may be necessary to maintain experimental stringency, it is also acknowledged that, first, arm cycling may be regarded as a novel task for some participants and, second, that by setting the cadence at 60 rpm, for example, participants may not be cycling at a preferred cadence. Taken together, these two factors may act to alter attentional demands, thus influencing the measures of corticospinal excitability.

When humans engage in a novel motor task, they typically focus on how to perform the said task, placing them in what is known as the “cognitive stage” of motor learning, according to the Fitts and Posner model [6]. This suggests that the level of cognitive effort, and thus in all likelihood the descending input, would be greater during this stage of learning. This is supported by work examining the time course of changes in corticospinal excitability when learning a novel motor task, albeit non-locomotor [7]. Holland et al. (2015) showed that the slope of the transcranial magnetic stimulation (TMS) evoked input/output (I/O) curve decreased as learning progressed, with the majority of the change occurring on the first of two training days. This suggests that as participants began the novel task, greater cognitive effort was required thus enhancing corticospinal excitability, an effect that decreased as the task lost its novelty.

Arm cycling is a motor task that may be considered novel and a number of studies have been published that have examined corticospinal excitability during cycling in humans [8,9,10,11,12,13,14]. Work from our lab has shown corticospinal excitability, assessed via TMS of the motor cortex projecting to the biceps brachii, to be higher during arm cycling in humans when the elbow was flexed (bottom dead centre) compared to an intensity- and position-matched tonic contraction [15]. This effect was due to enhanced supraspinal excitability, as there were no differences in the measures of spinal excitability. In that study, participants were required to maintain a pre-determined cadence (60 rpm) throughout the trial by observing their cadence on the ergometer monitor, and it was possible that this increased the attentional demands of the task. Research has shown that directed visual attention can induce an increase in neural activity in the fronto-parietal network, as evidenced in functional brain imaging studies [16]. It is thus possible that an increase in attention may increase corticospinal excitability during arm cycling, though we hypothesized that the difference was task-dependent and not simply due to the increased attentional demands of arm cycling [15].

Several studies have examined the influence of cycling cadence on neuromuscular activation. Marias et al. (2004) examined the effects of a spontaneous chosen crank rate (SCCR) and crank rates 20% higher and lower than the SCCR during arm cycling on integrated electromyography (iEMG) levels in the biceps brachii muscles in humans. The researchers concluded that there were no significant differences in the iEMG between the crank rate conditions of the biceps brachii, suggesting that the SCCR was not chosen to minimize the level of muscle activity and that the degree of muscle activation was similar between the two groups [17]. This finding is supported by research that showed no reduction in lower extremity muscle activation at a SCCR during leg cycling [18]. The iEMG assessed in these studies is a measure of the electrical activity in the muscle, representing the overall output of the motoneurone pool, and does not necessarily represent corticospinal excitability [8,13,14]. Therefore, it is unknown how a self-selected cadence (SSC) during arm cycling influences corticospinal excitability in comparison to a fixed cadence (FC).

The purpose of the current study was thus to determine if corticospinal excitability between SSC and FC arm cycling was different. It was hypothesized that corticospinal excitability, as assessed via the amplitude of motor evoked potentials (MEPs) elicited via TMS of the motor cortex, would not be different between SSC and FC arm cycling.

## 2. Materials and Methods

### 2.1. Ethical Approval

Prior to the experiment all participants were informed of the experimental protocol and written informed consent was obtained. This study was in accordance with the Helsinki declaration, and experimental procedures were approved by the Interdisciplinary Committee on Ethics in Human Research at the Memorial University of Newfoundland (ICEHR #20171250). All experimental procedures were in accordance with the Tri-Council guidelines in Canada, and potential risks of participation were disclosed to all participants.

### 2.2. Participants

Eleven participants (7 males and 4 females; 22 ± 2.14 years of age) were recruited from the School of Human Kinetics and Recreation (HKR) at Memorial University using a convenience sampling technique. Prior to testing, each participant completed a magnetic stimulation safety-checklist to screen for existing contraindications to magnetic stimulation (Rossi et al., 2009). To determine hand dominance, participants completed an Edinburgh Handedness Inventory questionnaire to ensure that all evoked responses were recorded from the dominant arm [19]. Additionally, to screen for existing contraindications to physical activity, each participant completed a Physical Activity Readiness Questionnaire (PAR-Q+) [20]. Participants were excluded if they had any neurological deficits or contraindications to magnetic stimulation or physical activity.

### 2.3. Experimental Set-Up

A one-group within-subjects design was used. Participants attended two lab sessions with at least 24 h in between visits. The first visit was for a half-hour familiarization session and the second was the testing session, lasting approximately 1 h. The experiment was completed on an arm cycle ergometer (SCIFIT ergometer, model PRO2 Total Body) with the arm cranks set at 180 degrees out of phase (see Figure 1). Each participant was advised to sit upright at a comfortable position from the arm cranks to ensure that they could maintain an upright posture throughout each cycling protocol. The seat height was adjusted to ensure the participant’s shoulders were in line with the centre of the arm shaft. The participants were informed to lightly grip the handles with their forearms in pronation. Each participant was required to wear wrist braces to limit wrist joint movement during cycling, to reduce the effects of the heteronymous reflex connections that exist between the wrist flexor muscles and the biceps brachii muscle [21].

All measurements were taken at a single position—6 o’clock relative to a clock face. This position was relative to the participants’ dominant hand, such that the TMS would be triggered when the right or left hand was at the 6 o’clock position for a right- or left-handed dominant individual, respectively. We have examined this position previously [8,9,10,11,12,13,15], as it corresponds to a period of high bicep brachii electromyography (EMG) activity during arm cycling since it occurs during mid-elbow flexion (i.e., movement from 3 o’clock to 9 o’clock).

The study required participants to cycle at two different cadences, both at a constant workload of 25 W. The cadences (FC and SSC) served as the independent variables in the study. The TMS and Erb’s point stimulation were delivered at the 6 o’clock position to elicit MEPs and maximal M-wave (M_max_) in the biceps brachii muscle in each condition. The MEP amplitude made relative to M_max_ and bEMG (background EMG; see below), as a measure of corticospinal excitability, served as the dependent variable. The SSC trial was completed first, followed by the FC trial, and responses were triggered as the arm crank of the dominant arm passed the 6 o’clock position.

### 2.4. Electromyography (EMG) Recordings

EMG activity was recorded from the biceps brachii and lateral head of the triceps brachii of the dominant arm using pairs of surface electrodes (Kendall^TM^ 130 conductive adhesive electrodes, Covidien IIC, Mansfield, MA, USA). The EMG was recorded using a bi-polar configuration with an interelectrode distance of 2 cm. Electrodes were placed in the middle of the muscle belly of the biceps brachii. A ground electrode was placed over the lateral epicondyle on the dominant arm. Prior to electrode placement, the skin at the recording site was shaved to remove hair, abraded using an abrasive pad to remove dead epithelial cells, and cleaned with an isopropyl alcohol swab to reduce impedance for the EMG recordings. Signals were sampled online at 5 kHz using a CED 1401 interface and Signal 5.11 software (Cambridge Electronic Design (CED) Ltd., Cambridge, UK). The EMG signals were amplified (gain of 300) and filtered using a 3-pole Butterworth band-pass filter (10–1000 Hz) using a CED 1902 amplifier.

### 2.5. Simulation Conditions

#### 2.5.1. Brachial Plexus Stimulation

Electrical stimulation of the brachial plexus at Erb’s point was used to measure M_max_ (maximal M-wave) (DS7AH, Digitimer Ltd., Welwyn Garden City, Hertfordshire, UK). The anode was placed on the acromion process and the cathode was placed over the skin in the supraclavicular fossa. A pulse duration of 200 μs was used and the stimulation intensity was gradually increased until the M-wave amplitude of the biceps brachii reached a plateau, referred to as M_max_. This stimulation intensity was increased by 10% and used for the remainder of the experiment to ensure maximal M-waves were elicited during each trial [22].

#### 2.5.2. Transcranial Magnetic Stimulation (TMS)

Motor evoked potentials (MEPs) were measured during both cycling trials from the biceps brachii and served as the dependent variable in the study. TMS (Magstim 200, Dyfed, UK) was used to elicit MEPs in the biceps brachii by placing a circular coil (13.5 cm outside diameter) over the vertex. TMS is a valid and reliable technique for eliciting MEPs, which are recorded from the muscle as a measure of the excitability of the corticospinal tract (Rothwell et al., 1991). The vertex was located by measuring the mid-point between the nasion and the inion and between the participant’s tragi, and marks were placed for both measurements directly on the scalp. The intersection of the measurements was defined as the vertex [13,15,23,24]. The same researcher held the coil for each trial and was vigilant in ensuring that the coil was held parallel to the floor and remained aligned with the vertex throughout each trial. The current preferentially activated the right or left motor cortex, depending on hand dominance. The stimulation intensity was set during cycling (60 rpm and 25W) with MEPs evoked when the dominant hand was at the 6 o’clock position. The stimulus intensity was measured as a percentage of the maximum stimulator output (MSO), and the intensity was increased until the participant’s active motor threshold (AMT) was found. The AMT was defined as the lowest stimulus intensity required to evoke 5 clearly discernable MEPs (~200 μV) in 10 trials during cycling. Once the AMT was found, the MSO was increased by 10% to ensure that clearly discernable MEPs were recorded, and this stimulation intensity was then used for all trials.

#### 2.5.3. Experimental Protocol

After the stimulation intensities were set for the TMS and Erb’s point stimulation, the cycling trials were completed. The participant was first instructed to cycle forward at a comfortable pace, and the monitor displaying the cycling cadence was moved out of the participant’s sight, such that the participant was blinded to their cycling cadence. When the participant reached a steady cadence, as observed by the researcher, the trial was started. A steady cadence was defined as a cadence that fluctuated no more than ±1 rpm over a 5 s period. While the participant was cycling, the researcher recorded the cadence every 5 s and calculated the average cadence over the duration of the trial. After a 1 min break the participant was instructed to cycle forward maintaining a target cadence, as specified by the researcher, by observing their cadence on the monitor. This target cadence (FC) was equal to the average of the cadence over the duration of the SSC trial. During both trials the arm ergometer was set to a fixed power output of 25 W. While cycling, each participant received 12 MEPs and 2 M-waves per trial, which were delivered when the dominant hand passed the 6 o’clock position. The order of the stimulations was randomized during the trial, and the stimulations were evoked every 7–8 s. To prevent anticipation of the stimulation, 2 frames without stimulation were added. The total length of cycling was approximately 2 min per trial.

#### 2.5.4. Data Analysis

Data were analyzed off-line using Signal 5.11 software (Cambridge Electronic Design Ltd., Cambridge, UK). To determine if the central motor drive projecting to the biceps brachii was similar between the two arm cycling conditions, the mean rectified EMG 50 ms prior to the TMS stimulus artifact was measured [15]. The peak-to-peak amplitude of all evoked responses (MEP and M-wave) were measured from the initial deflection of the voltage trace from background EMG to the return of the trace to the baseline level. MEP amplitudes can change as a result of changes to M_max_, thus MEPs were normalized to M_max_ evoked during the same trial to account for potential changes in peripheral excitability. All measurements were taken from the averaged files of all 12 MEPs and 2 M-waves. All measurements were made from the dominant arm.

#### 2.5.5. Statistical Analysis

To compare the pre-stimulus EMG between the conditions (SSC and FC), paired-samples *t*-tests were used. Additionally, paired-samples *t*-tests were used to assess whether statistically significant differences in MEP amplitudes normalized to M_max_ occurred between the SSC and FC conditions. All statistics were completed on group data with a significance level of *p* < 0.05. All data are reported as mean ± SE (standard error) in the figures.

## 3. Results

***Cycling cadence***. Figure 2 shows the group mean cycling cadence in revolutions per minute (rpm) during the SSC and FC arm cycling trials. The cycling cadences for each condition were not significantly different (mean cadence—SSC was 62 ± 6.4 rpm and FC was 63 ± 6.9 rpm; *p* = 0.118).

***MEP amplitude***. Figure 3A shows representative data for the MEP amplitudes from one participant for both the SSC and FC cycling conditions. Figure 3B shows the group mean MEP amplitudes expressed as a percentage of M_max_ of the biceps brachii during the SSC and FC arm cycling trials. The average MEP amplitude (normalized/standardized to M_max_) when cycling at a SSC and a FC was 16.2% (SD (standard deviation) = 12.25) and 14.1% (SD = 11.75), respectively, with a mean difference of 2.1%. This difference was not statistically significant (*p* = 0.146).

***Pre-stimulus EMG of the biceps brachii for MEPs***. The group mean (*n* = 11) pre-stimulus EMG of the biceps brachii prior to the TMS stimulus artifact during the SSC and FC arm cycling can be seen in Figure 4. As a group, the mean pre-stimulus EMG for the SSC and FC arm cycling trials was 30.2 ± 4.58 μV and 32.1 ± 5.82 μV, respectively. There was no significant difference between the values (*p* = 0.061).

***Pre-stimulus EMG of the triceps brachii for MEPs***. The group mean (*n* = 11) pre-stimulus EMG of the triceps brachii prior to the TMS stimulus artifact during the SSC and FC arm cycling can be seen in Figure 5. As a group, the mean pre-stimulus EMG for the SSC and FC arm cycling trials was 8.9 ± 2.12 μV and 9.4 ± 2.68 μV, respectively. There was no significant difference between the values (*p* = 0.58).

## 4. Discussion

This is the first study to compare corticospinal excitability projecting to the biceps brachii between self-selected (SSC) and fixed cadence (FC) arm cycling. There were no significant differences in corticospinal excitability, as assessed via TMS-evoked MEP amplitudes recorded from the biceps brachii, between the two arm cycling conditions. Maintaining a pre-determined cadence (FC) during arm cycling does not increase corticospinal excitability, when compared to cycling at a voluntarily chosen cadence (SSC).

A prior concern in studies from our lab and also in the work of others was that the attentional demands of maintaining a set cadence could inadvertently alter (likely increase) the measures of corticospinal excitability. The current finding that corticospinal excitability is not different between SSC and FC arm cycling lends support to our previous finding that corticospinal excitability is task-dependent and is higher during arm cycling than an intensity- and position-matched tonic contraction [15]. In that study, the participants were required to maintain a pre-determined cadence (60 rpm) while arm cycling rather than a voluntarily chosen cadence [15]. Thus, it was unknown if the increase in supraspinal excitability projecting the biceps brachii at the 6 o’clock position was due to the arm cycling task or if it resulted from a greater attentional demand to maintain the set cadence. The results from the current study indicate that focusing on maintaining a fixed cadence does not increase the overall excitability of the corticospinal tract, compared to arm cycling at a SSC. Thus, the increase in corticospinal excitability during arm cycling that we reported was likely to be task-dependent and not attributable to the fact that the participants had to focus on maintaining a cadence of 60 rpm [15]. This is indirectly supported by prior work assessing the EMG of both arm and leg muscles during either arm [17] or leg [18] cycling, respectively. In the aforementioned studies, there was no influence of the SSC or the FC on EMG amplitudes, though there were no measures of corticospinal excitability.

### 4.1. Attentional Focus and Corticospinal Excitability

Prior work has shown that visual attention modulates corticospinal excitability and directing visual attention toward the specific features of an observed action facilitates corticospinal excitability more than passive observation [25,26]. Attention can be directed to highly salient stimuli based on their physical properties (e.g., brightness, colour, and speed) or toward stimuli that are important for one’s current task [27]. In this study during the FC condition, participants were instructed to focus on the monitor that displayed the cadence they were cycling at and were instructed to maintain a set cadence and speed up or slow down based on the observed cadence. In contrast, during the SSC condition participants were not able to see the monitor and were not instructed to focus on any particular object in the external environment. Although participants were instructed to focus on the cadence on the monitor throughout the FC trial, corticospinal excitability projecting to the biceps brachii was not increased when compared to the SSC trial. A possible explanation for the lack of increase in corticospinal excitability during the FC trial is that it is unknown if the participant maintained their focus on the cadence displayed on the monitor throughout the entire trial, as eye tracking devices were not used. In addition, much of the literature regarding increases in corticospinal excitability with focused attention has been on the observation of human movement and the activity in the putative mirror neuron system. Notably, corticospinal excitability is facilitated during action observation and more so during goal-directed actions (e.g., grasping an object) when attention is directed to task-relevant features of the observed action [28]. In this study, the participants were not observing an action but were rather observing numbers on a monitor that were relevant to their behavioural goal (maintaining a set cadence). Thus, the theory that corticospinal excitability is facilitated during action observation due to the increased activity in the mirror neuron system may not apply in the present study.

### 4.2. Methodological Considerations

Additional factors should be considered when interpreting the present results. This study assessed MEP amplitudes and therefore conclusions can only be made regarding the overall excitability of the corticospinal tract. In future, research assessing spinal excitability, with TMES (transmastoid electrical stimulation) for example, to the target muscle to determine if changes in corticospinal excitability are occurring at the spinal and/or supraspinal level may be of interest [29]. For instance, it is possible that supraspinal excitability increased during the FC trial, and the increase was masked by a reduction in spinal excitability, resulting in no change in the overall excitability of the corticospinal tract. In order to decipher between supraspinal and spinal excitability both TMES and TMS need to be used. The reason we chose the 6 o’clock position, however, was because in our prior work we have shown that corticospinal excitability is higher during arm cycling than a tonic contraction at that position while spinal excitability is not. Thus, it is unlikely that spinal excitability was different in the present study. Additionally, some participants in this study had previous experience with arm cycling and therefore may have required less attentional focus to execute the task. However, we purposely included a familiarization session for all participants to minimize this threat to internal validity by allowing participants to practice arm cycling.

## 5. Conclusions

The novel finding in this study is that corticospinal excitability, as assessed by changes in MEP amplitude, projecting to the biceps brachii is not different between SSC and FC arm cycling. We can indirectly (because attention was not directly measured) conclude that corticospinal excitability during arm cycling is independent of attentional demands, as corticospinal excitability is not different when focusing attention on maintaining a set cadence compared to cycling at a voluntarily chosen cadence.

## Figures and Tables

**Figure 1 brainsci-09-00041-f001:**
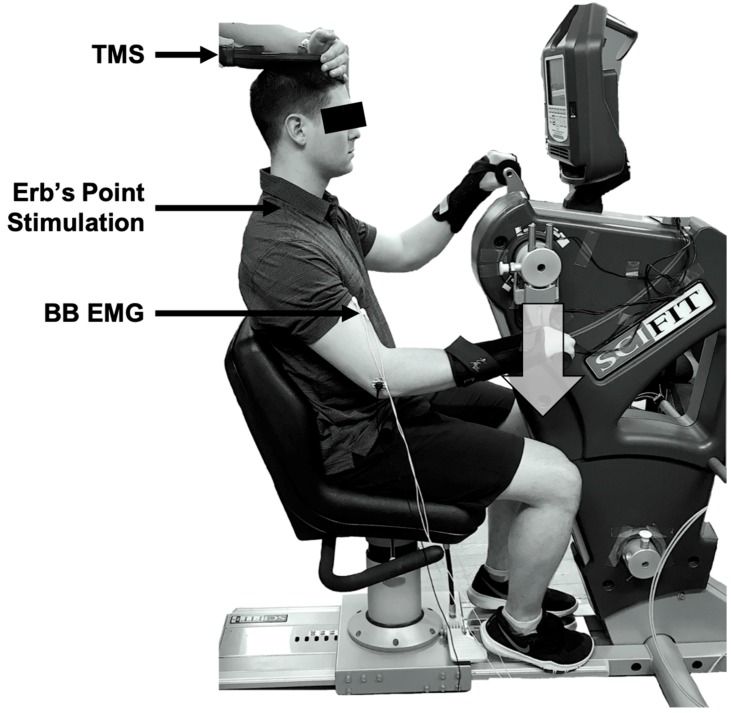
Experimental set-up. Arm cycling was performed in the forward direction, with stimulations occurring when the dominant arm passed the 6 o’clock position (i.e., bottom dead centre) when the biceps brachii was active. This position is denoted by the large, grey downwards arrow. TMS = transcranial magnetic stimulation; BB = biceps brachii; EMG = electromyography.

**Figure 2 brainsci-09-00041-f002:**
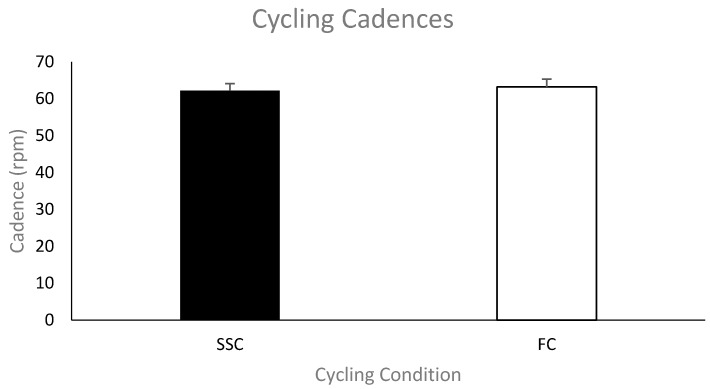
Mean cycling cadences for each group (SSC = black and FC = white). Data (*n* = 11) are shown as mean ± SE (standard error).

**Figure 3 brainsci-09-00041-f003:**
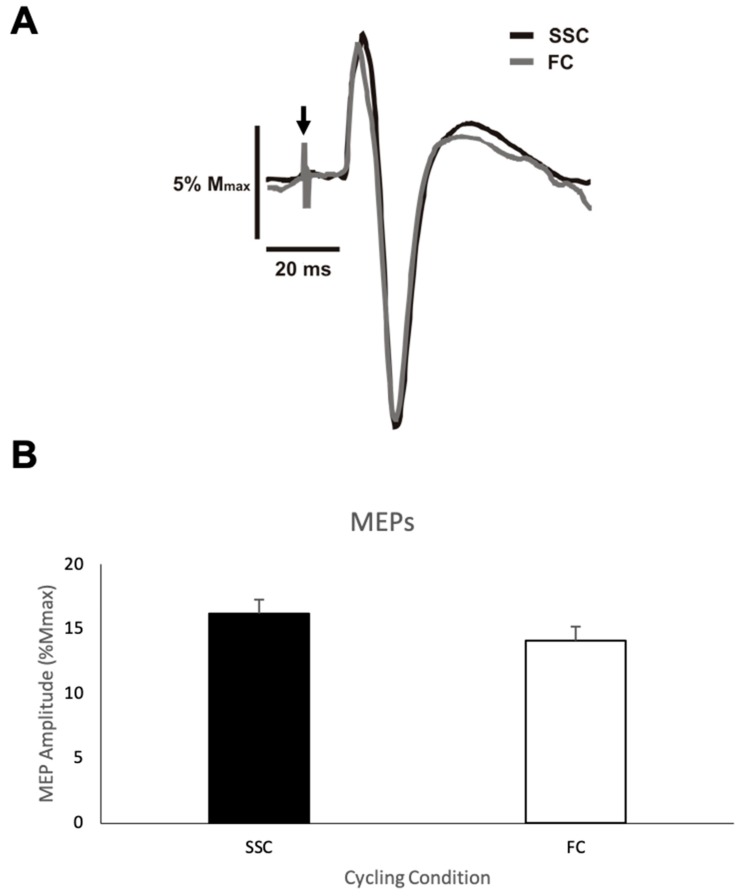
(**A**) Representative motor evoked potential (MEP) amplitudes from one participant for each cycling condition (SSC = black and FC = grey). Downward arrow indicates the location of the stimulus artifacts that have been adjusted in size for figure clarity. (**B**) Mean transcranial magnetic stimulation (TMS) evoked MEP amplitudes as a percentage of the maximal M-wave (Mmax) for each group (SSC = black and FC = white). Data (*n* = 11) are shown as mean ± SE.

**Figure 4 brainsci-09-00041-f004:**
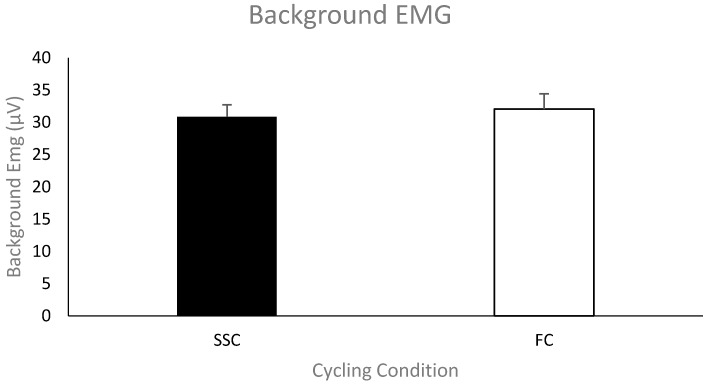
Mean of the average rectified electromyography (EMG) amplitude for the biceps brachii prior to TMS-evoked MEPs for each group (SSC = black and FC = white). Data (*n* = 11) are shown as mean ± SE.

**Figure 5 brainsci-09-00041-f005:**
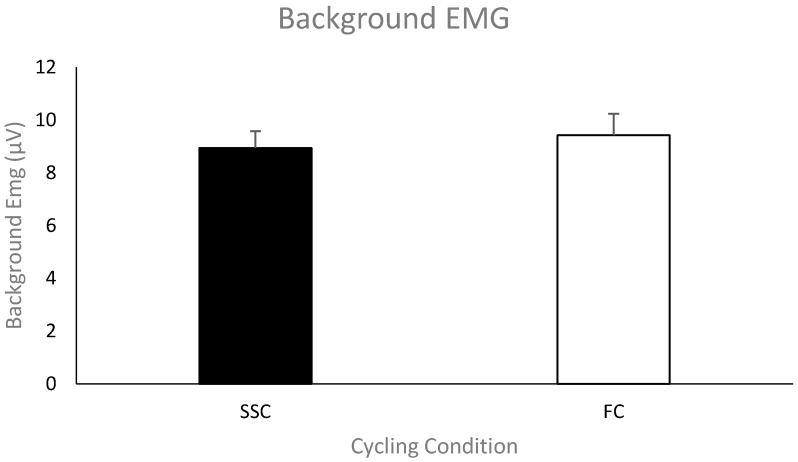
Mean of the average rectified EMG amplitude for the triceps brachii prior to TMS-evoked MEPs for each group (SSC = black and FC = white). Data (*n* = 11) are shown as mean ± SE.

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
