# Peer review of "Corticospinal Excitability to the Biceps Brachii is Not Different When Arm Cycling at a Self-Selected or Fixed Cadence"

_brainsci, 2019, doi:10.3390/brainsci9020041_

Reviewer 1 Report

Please see PDF attached 

Author Response

We wish to thank each of the reviewers for their suggestions and comments. We sincerely appreciate the time and effort you have invested in our research. We address each concern below.

Reviewer #1 (Comments to the Author):

Comment:Suggest at the authors discretion further inclusion of recent detail on the following two neurology and neuro mechanics themes to enhance the overall rationale and discussion sections. I hope this is helpful.

Response:Thank you for the time you have taken to provide us with this insightful information.

Comment:Lines 24 – 37. Central Pattern Generators 

Central pattern generators are complex structures for which many of the cellular elements have not yet been unraveled. Nonetheless, compelling evidence supports key roles in controlling biological rhythms such as locomotion in most if not all vertebrate species and that current working hypotheses implicate CPG’s in the control of rhythmic movement probably localized between the human cervical and lumbar spinal cord. The implication being that the characteristics of rhythmic arm movement should be somewhat similar to that of rhythmic leg movement. In the context if this study this general hypothesis has been applied to the study of regulation of reflex patterns in arm muscles during rhythmic arm cycling.

Response: We appreciate the reviewer’s understanding of CPGs and the working hypothesis that we apply to our work during arm cycling. The reviewer is correct in that we utilize arm cycling as a model of a CPG-mediated motor output, as has been done during leg cycling and human locomotion. For clarity, however, we wish to point out that we do not study the regulation of reflex patterns during arm cycling. Instead, we use non-invasive stimulation techniques to examine the modulation of the output of the motor system (i.e. the corticospinal pathway) during arm cycling, which are quite different. 

Comment: Lines 38-76 Similarly it would be helpful to include that in the In the process of human arm motion, neural subsystems such as the cortex (cellular output layers betz cells etc.), basal ganglia and cerebellum output modulates descending effects on the and the spinal cord. Consequently it is generally acknowledged that all of these subsystems play different roles in human arm motion.

Response: This is a great point that the reviewer makes and one that we acknowledge may be true. Indeed we have discussed the role of the cerebellum in recent years given the method recently published by Bernadette Murphy’s group to assess cerebellar modulation of motor cortical output (Baarbe et al. 2014). Having said that we have not examined the role of these ‘other’ supraspinal structures in the modulation of corticospinal excitability during arm cycling. While these other brain structures may indeed play a role in modulating corticospinal excitability during arm cycling, we feel as though they are outside the scope of the present manuscript.

Comment:(Interesting article) 

“Arm motion control model based on central pattern generators” 

Zhigang ZHENG, Rubin WANG† Institute for Cognitive Neurodynamics, East China University of Science and Technology, Shanghai 200237, China 

Appl. Math. Mech. -Engl. Ed., 38(9), 1247–1256 (2017) DOI 10.1007/s10483-017-2240-8 

Shanghai University and Springer-Verlag Berlin Heidelberg 2017

Response: Thank you for this paper. We will certainly think about including this as a reference in our future work.

Comment: The Methodology is clearly and very nicely explained and contextual in development to previous study designs:

Response:Thank you for your kind words.

Comment:Lines 113 – 119suggest to demonstrate this approach and overall set up an image be included – and for the non-initiated reader the 6.o clock position needs some clarification.

Response: We agree and have added an experimental setup figure (Figure 1), which displays the 6 o’clock position.

Figure 1. Experimental setup. Arm cycling was performed in the forward direction, with stimulations occurring when the dominant arm passed the 6 o’clock position (i.e. bottom-dead centre) when the biceps brachii is active. This position is denoted by the large grey downwards arrow.

Comment:Similar with regard to Erbs point

Magnetic resonance imaging has formalized anatomical analysis of the compartments of the brachial plexus – it would be helpful to define the locus of Erbs point and state its significance as a neural generator and standardized locus adopted in th se study design’s.

Lines 134 -140 Electrical stimulation of the brachial plexus at Erb’s point was used to measure Mmax (maximal 134 M-wave) (DS7AH, Digitimer Ltd., Welwyn Garden City, Hertfordshire, UK). The anode was placed 135 on the acromion process and the cathode was placed over the skin in the supraclavicular fossa. A 136 pulse duration of 200 s was utilized and the stimulation intensity was gradually increased until the 137 M-wave amplitude of the biceps brachii reached a plateau, referred to as

Response: We appreciate the interesting comments regarding MRI analysis of the brachial plexus. The use of electrical stimulation of the brachial plexus at Erb’s point is commonly used to examine changes in peripheral excitability to muscles of the upper limb, including biceps, in research for numerous years by several laboratories, including the Gandevia lab (Butler et al., 2003, Petersen et al., 2002).Unfortunately, we currently have no way to define the specific locus of the stimulation other than by providing the bony landmarks, which we have done as in our previous works (Copithorne et al. 2015; Forman et al. 2014; Forman et al. 2018; Forman et al. 2015; 2016a; Forman et al. 2016b; Lockyer et al. 2018; Pearcey et al. 2014; Spence et al. 2016). In addition, we did not have any issues evoking maximal M-waves in any of our participants, which ultimately were then used to normalize our responses from TMS of the motor cortex to. Thus, this should not influence the current findings.

Comment: MRI anatomic analysis of the brachial plexus has benefited significantly from neuro graphic sequences. 

The anatomical points used as reference to assess MRN images are the clavicle, the first rib, the subclavian artery and vein and the scalene anterior and medius muscles. 

The brachial plexus is formed from the primary ventral branches of the spinal nerves that originate in the cervical (C) 5, 6, 7, 8 and thoracic (T) segments 

1. In some individuals smaller ventral branches C4 and T2 participate.13 

2. Note this is the proximal compartment with ERBs point localized between the margins of the proximal and middle compartments

Response:Thank you for this information and the detailed images as well. We appreciate these comments and will consider them in our future works. See above response regarding the use of Erb’s point stimulation.

Reviewer 2 Report

In this paper, the authors investigated cortical excitability during rhythmic movements of healthy volunteers. The subjects were asked to maintain a given arm cycling frequency, or to cycle at a self-selected frequency. Cortical excitability was assessed through motor evoked potentials of the biceps in both the conditions. The results indicate that this attentional demand does not influence this measure of cortical excitability.

Several physical therapeutic exercises are used in rehabilitation programs and arm cycling is one of them. However, there is not enough data to show whether these tasks really exert an effect on neural systems. Therefore, the findings of the present paper can be of interest for a wide readership, even if the attentional demand induced no additional effect. The paper is clear, concise and well written. The gathered data were discussed fairly. The methodology is adequate to draw the proposed conclusions.

I have just some minor remarks:

The paper should include a figure of motor evoked potential recorded during cycling activity. It would be useful for readers unfamiliar with this technique.

The authors state that “corticospinal excitability is task-dependent and is higher during arm cycling than an intensity- and position-matched tonic contraction”. A tonic contraction is a very different type of muscle activity compared to cycling. Indeed, tonic contraction very likely does not involve the activation of central pattern generators. Have any other motor patterns been tested with this MEP model? For example, the use of a rowing machine or an elliptical trainer? The authors are invited to briefly discuss this topic.

Author Response

Reviewer #2 (Comments to the Author):
Comment:In this paper, the authors investigated cortical excitability during rhythmic movements of healthy volunteers. The subjects were asked to maintain a given arm cycling frequency, or to cycle at a self-selected frequency. Cortical excitability was assessed through motor evoked potentials of the biceps in both the conditions. The results indicate that this attentional demand does not influence this measure of cortical excitability.

Several physical therapeutic exercises are used in rehabilitation programs and arm cycling is one of them. However, there is not enough data to show whether these tasks really exert an effect on neural systems. Therefore, the findings of the present paper can be of interest for a wide readership, even if the attentional demand induced no additional effect. The paper is clear, concise and well written. The gathered data were discussed fairly. The methodology is adequate to draw the proposed conclusions.

Response:First, let us say thank you for your kind words. 

Comment: I have just some minor remarks:

The paper should include a figure of motor evoked potential recorded during cycling activity. It would be useful for readers unfamiliar with this technique.

Response: Thank you for this comment. We have added a figure of raw MEPs recorded during arm cycling from both the SSC and FC conditions for clarity (Figure 3A).

Figure 3. (A) Representative MEP amplitudes from one participant for each cycling condition (SSC = black and FC = grey). Downward arrow indicates location of stimulus artefacts that have been adjusted in size for figure clarity. (B)Mean TMS evoked MEP amplitudes as a percentage of Mmax for each group (SSC = black and FC = white). Data (n=11) is shown as mean ±SE.

Comment: The authors state that “corticospinal excitability is task-dependent and is higher during arm cycling than an intensity- and position-matched tonic contraction”. A tonic contraction is a very different type of muscle activity compared to cycling. Indeed, tonic contraction very likely does not involve the activation of central pattern generators. Have any other motor patterns been tested with this MEP model? For example, the use of a rowing machine or an elliptical trainer? The authors are invited to briefly discuss this topic.

Response: The majority of the research done regarding changes in corticospinal excitability (via TMS-evoked MEPs) during human movement has been assessed during tonic contractions (Martin et al., 2006; Oya et al., 2008; Pearcey et al., 2014). More recently, however, corticospinal excitability has been assessed during rhythmic locomotor-like movements, such as walking (Barthelemy et al., 2010; Capaday et al., 1999) and leg cycling (Sidhu et al., 2012; Weavil et al., 2015). Our lab has also published extensively on rhythmic arm cycling (see reference list below) and we have even compared cycling to an intensity matched tonic contraction, which as the reviewer points out are two very different motor outputs as shown via differences in corticospinal excitability. To our knowledge, corticospinal excitability has not been measured during the movement on a rowing or an elliptical trainer machine. However, this could be worth investigating. We have recently acquired a SCIFIT recumbent stepper machine (similar to an elliptical but in a recumbent position) for our laboratory and aim to assess corticospinal excitability during the movement in the future.

We wish to thank each of the reviewers for their suggestions and comments. We sincerely appreciate the time and effort you have invested in our research. We address each concern below.

Reviewer 3 Report

The present study compared corticospinal excitability to the biceps brachii muscle during arm cycling at a self-selected and a fixed cadenc. The corticospinal excitability were found to be not different between groups. The manuscript is well written and the findings support conclusions. However the paper should be checked thoroughly for some English grammar and spelling errors.

Author Response

Reviewer #3 (Comments to the Author):
Comment: The present study compared corticospinal excitability to the biceps brachii muscle during arm cycling at a self-selected and a fixed cadenc. The corticospinal excitability were found to be not different between groups. The manuscript is well written and the findings support conclusions. However the paper should be checked thoroughly for some English grammar and spelling errors.

Response:First of all, we would like to thank you for your kind comments. We have read through the manuscript and checked again for English and grammar errors and have corrected them where deemed necessary.